**Subject Category:**
Biology (whole organism)

behaviour/ecology

larval nutrition, reproduction, trans-generational effects, development, density, aggregation

**Author for correspondence:**
Juliano Morimoto
e-mail: juliano.morimoto@mq.edu.au

# Larval foraging decisions in competitive heterogeneous environments accommodate diets that support egg-to-adult development in a polyphagous fly

Juliano Morimoto[1,2], Shabnam Tarahi Tabrizi[1], Ida Lundbäck[1], Bishwo Mainali[1], Phillip W. Taylor[1] and Fleur Ponton[1]

[1]Department of Biological Sciences, Macquarie University, New South Wales 2109, Australia
[2]Programa de Pós-Graduação em Ecologia e Conservação, Federal University of Paraná, Curitiba 19031, CEP: 81531-990, Brazil

JM, 0000-0003-3561-1920

In holometabolous insects, larval nutrition is a key factor underpinning development and fitness. Heterogeneity in the nutritional environment and larval competition can force larvae to forage in suboptimal diets, with potential downstream fitness effects. Little is known about how larvae respond to competitive heterogeneous environments, and whether variation in these responses affects current and next generations. Here, we designed nutritionally heterogeneous foraging arenas by modifying nutrient concentration, where groups of the polyphagous fruit fly *Bactrocera tryoni* could forage freely at various levels of larval competition. Larval foraging preferences were highly consistent and independent of larval competition, with greatest foraging propensity for high (100%) followed by intermediate (80% and 60%) nutrient concentration diets, and avoidance of lower concentration diets (less than 60%). We then used these larval preferences (i.e. 100%, 80% and 60% diets) in fitness assays in which larvae competition was maintained constant, and showed that nutrient concentrations selected by the larvae in the foraging trials had no effect on fitness-related traits such as egg hatching and pupation success, adult flight ability, sex ratio, percentage of emergence, nor on adult cold tolerance, fecundity and next-generation pupal weight. These results

support the idea that polyphagous species can exploit diverse hosts and nutritional conditions with minimal fitness costs to thrive in new environments.

## 1. Introduction

In holometabolous insects, nutritional stress at the larval stage can have long-lasting fitness implications [1–4]. Poor larval diet has been shown to result in low adult body mass and fecundity for many insect species including some Lepidoptera, Coleoptera and Diptera [5–13]. For instance, the reproductive rate of ladybird beetles (*Harmonia axyridis*) is consistently lower when larvae have experienced poor nutritional environments, even if nutrition is abundant in adulthood [13]. In mosquitoes, nutritional deprivation at the larval stage increases developmental time [14] and results in adults with smaller body size and elevated sensitivity to toxins [15,16]. In *Drosophila melanogaster*, high larval crowding or food dilution results in adults with lower body mass, lower reproductive potential and, in some cases, next-generation offspring with lower body mass [4,17–19].

The larvae of some polyphagous insects display remarkably precise dietary choices that promote development [6,7,20–22]. For example, when given a choice between chemically defined diets, *Ceratitis capitata* (Mediterranean fruit fly, or 'medfly') larvae regulate nutrient intake to promote larval growth and adult size [6,7]. Some *Drosophila* species are able to select diets that minimize development time [22,23]. These studies indicate that larvae can respond to variation in nutritional conditions to maximize individual development. However, a potential caveat can arise when studies focus solely on the average dietary choices of foraging individuals rather than assessing effects across the full diversity of accepted diets, without considering how larval competition modulates larval foraging preferences (e.g. [7,21,22,24,25]). Larvae commonly forage in aggregations which may vary in size, both because females oviposit eggs in batches and because larvae may have a common affinity for a particular location [21,26–28]. This creates the potential for larval competition and socially dependent larval dietary choices. Moreover, the nutrients in plants and fruits are not evenly distributed, meaning that larval foraging substrates tend to be heterogeneous (e.g. [22,29–31]). *Ceratitis capitata* larvae move to more nutritious parts of the food resource, which suggests that heterogeneity in the nutritional environment is an important factor in larval foraging behaviour [32]. Therefore, larval competition and the heterogeneity in the nutritional environment can affect the average as well as the variation in larval choices, which in turn can affect individual fitness [33]. Despite this, feeding decisions of larvae foraging in aggregations and in heterogeneous nutritional environments, as well as the consequences of such decisions for larval development and fitness-related traits in adulthood, have been largely overlooked. Fitness costs of foraging in competitive heterogeneous nutritional environments can arise in many ways. For instance, individuals foraging in competitive heterogeneous nutritional environments might be forced to move around more often and/or feed on suboptimal diets. If suboptimal feeding is sustained throughout development, this can lead to downstream effects on developmental time, body size and tolerance to environmental stresses, and fecundity [23,34–36], with unknown trans-generational effects. However, there has been no detailed investigation of the behavioural responses and potential costs associated with foraging in competitive heterogeneous nutritional environment. Key questions remain unanswered, such as 'How do individuals adjust their foraging behaviour in competitive heterogeneous environments?'; and 'What are the potential fitness costs of foraging choices made in heterogeneous environments?'. The answer to these questions can help us better understand how polyphagous insect larvae forage under different competition levels and in nutritionally heterogeneous environments, and how larval foraging patterns can support individual development and adult performance in changing environmental conditions.

In the present study, we explored these questions by investigating the foraging patterns of larvae of a polyphagous fruit fly species at different larval competition levels and in heterogeneous nutritional environments, as well as how foraging patterns could affect adult and next-generation fitness traits. To do this, *Bactrocera tryoni* Froggatt (Tephritidae) larvae were assembled in different densities and were allowed to forage freely in a multi-choice foraging assay where patches varied in nutrient concentration from high (100%) to low (20%). Next, we provided either one of the three most preferred diets from the larval choice experiment and allowed egg-to-adult development in controlled larval competition levels, while quantifying fitness traits including pupal recovery (i.e. percentage of pupae obtained relative to the total number of eggs that hatched), daily pupal recovery (i.e. the daily number of pupae relative to the total number of pupae obtained), the percentage of adult emergence from pupae, adult flight ability, cold tolerance and adult fecundity, as well as next-generation pupal

weight. Our findings shed light on the resilience of a polyphagous fruit fly to heterogeneity in the nutritional environment, which can provide insights into how other polyphagous species cope with unpredictable nutritional environments.

# 2. Predictions

Our predictions were as following:

(1) Based on previous studies [6,7,20–22], we predicted that larvae would prefer the diet with the highest nutrient concentration when given a choice among foraging patches.
(2) Because it is unclear how larval competition affects larval foraging behaviour, we predicted that larval foraging choices would be similar across all larval densities.
(3) Based on the literature (e.g. [4,17–19]), diets with lower nutrient concentration would reduce adult body mass and fecundity.
(4) Formulation of predictions for the effects of nutrient concentration on flight ability and stress tolerance are largely speculative due to the lack of previous studies in the field. We therefore stipulated null predictions that nutrient concentration would not affect adult flight ability and cold stress tolerance.

# 3. Material and methods

## 3.1. Fly stock and egg collection

We collected eggs from a laboratory-adapted stock of *B. tryoni*. The colony was established in 2015 (more than 17 generations old) and has been maintained in non-overlapping generations in humidity ($65 \pm 5\%$) and temperature ($25 \pm 0.5°C$) controlled rooms with light cycles of 12 h light: 0.5 h dusk: 11 h dark: 0.5 h dawn). Adults were allowed to feed freely from separate dishes of hydrolysed yeast (MP Biomedicals Cat. no. 02103304) and commercial refined cane sugar (CSR® White Sugar), while larvae were maintained using the Chang-2006 diet formulation of Moadeli *et al.* [37] for the last seven generations. Eggs were collected for 2 h in a 300 ml semi-transparent white soft plastic bottle (low-density polyethylene) that had numerous perforations of less than 1 mm diameter through which females could oviposit. To maintain humidity, the bottle contained 20 ml of water. Approximately 3500 eggs (250 µl of eggs suspended in water) were transferred to 150 ml of larval diet with a pipette. Eggs were allowed to hatch and larvae to develop until they reached second instars, after which the larval diet choice experiment began.

## 3.2. Experimental design

### 3.2.1. Experiment 1: larval diet choice

Five experimental diets that varied in nutrient concentration were designed: the control and reference 100% Chang-2006 diet, which has proven effective for rearing larvae of this species [37], followed by diets with 80%, 60%, 40% and 20% nutrient concentration relative to the control diet (electronic supplementary material, table S1). Twenty millilitres of freshly prepared diet was poured into 90 mm diameter Petri dishes and allowed to set. We also prepared an agar solution that contained the same components as the experimental diets except that no yeast or sugar was included. Twenty millilitres of the agar solution was used to cover the base of additional Petri dishes that then served as 'foraging arenas'. The pH of all experimental diets and the agar base was adjusted to 3.8–4 with citric acid. For the experimental diets used in the foraging experiment, Brewer's yeast and sucrose were obtained from MP Biomedicals (Cat. no. 02903312 and 02902978, respectively), Nipagin was obtained from Southern Biological (Cat. no. MC11.2), and all other chemicals were obtained from Sigma Aldrich®. After setting, five equally spaced discs were excised from the agar base of each foraging arena by perforating the agar with a 25 mm diameter plastic tube. The same tube was used to excise discs of diet from the Petri dishes of experimental diets. The discs of experimental diets were then deposited in the holes in the agar base of the foraging arenas (see electronic

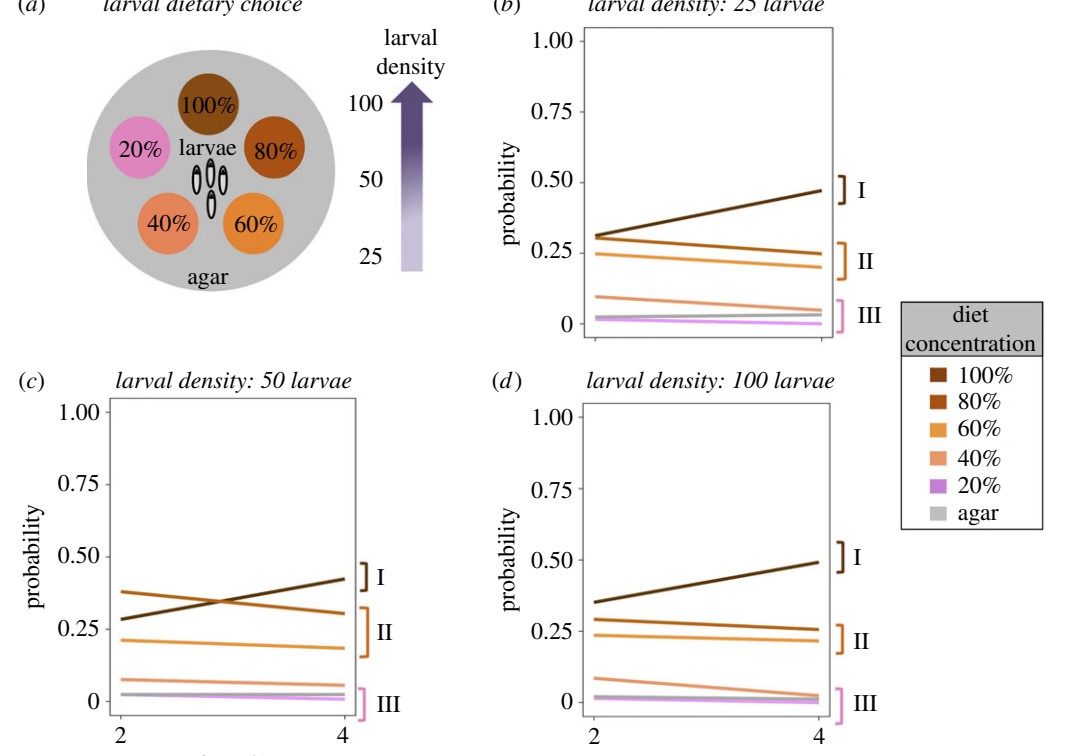

**Figure 1.** Patterns of larval foraging behaviour result can be ranked into preferred, intermediate and unwanted diet choices. (a) Schematic of our experimental set-up for the larvae dietary choices. The circular foraging arena was used to assess larval foraging decisions in regard to nutrient concentration. (b–d) Probability of larvae choosing each of the five diets when the density of individuals in the foraging group was 25 (b), 50 (c) and 100 (d) larvae. I, preferred diet (100% macronutrient concentration); II, intermediate diets (80% and 60% macronutrient concentration); III, unwanted diets (less than 60% macronutrient concentration).

supplementary material, figure S1). Because the agar solution did not contain macronutrients but contained other nutrients, the agar base was considered as a foraging option (see electronic supplementary material, table S1 and 'Statistical analyses' below). Thus, larvae had a total of six options (i.e. five experimental diets + agar). We placed 25, 50 or 100 larvae at the centre of each foraging arena (figure 1a), which allowed us to assess whether the density of conspecifics affected the patterns of larval foraging behaviour. We had five replicates foraging arenas per larval density ($n = 15$). Numbers of larvae were counted on each diet patch and on the agar base 2 h and 4 h after larvae were placed on the arena.

### 3.2.2. Experiment 2: performance

We assessed how nutritional variation in the diets that were accepted by the larvae in the foraging experiment influences individual performance. Diets that were within the range accepted by larvae in Experiment 1 were selected, namely diets with 100% nutrient concentration, which was occupied by the greatest proportion of larvae in Experiment 1 (preferred diet), as well as 80%, and 60% nutrient concentrations (henceforth referred to as 'diet treatment'), which were occupied by an intermediate number of larvae (intermediate diets; figure 1 and electronic supplementary material, figure S1). Diets were prepared as described in electronic supplementary material, table S1. To measure developmental traits and adult emergence, groups of 100 eggs were placed with a soft brush on black filter paper at the centre of 90 mm Petri dishes that contained 25 ml of one of the three diet treatments (i.e. 100%, 80% and 60% experimental diets; $n = 18$). The number of eggs that hatched was scored after 3 days (egg hatching success), after which the filter paper was removed. The larvae were then left to develop until they reached their third instar (6 days after the onset of the experiment), and Petri dishes were then placed in 1 l plastic containers that contained *ca* 30 g of fine vermiculate for pupation. Pupae were sifted from the vermiculite and counted at 7, 8 and 9 days after eggs were placed on the diets.

'Pupal recovery' was measured as the total number of pupae recovered 9 days after the eggs were placed in the diets, divided by the number of eggs that hatched in each diet treatment multiplied by 100 (results in %). 'Daily pupal recovery' was calculated as the number of pupae obtained each day divided by the total number of pupae obtained per diet treatment, multiplied by 100 (results in %). No larvae remained in the diets after 9 days since eggs were placed in the diets. For the first 2 days of pupation, 10 pupae per diet treatment per replicate per day ($n = 360$) were sampled and weighed 5 days after pupation using a Sartorius® ME5 balance (0.0001 g precision) as a measure of 'pupal weight'. To obtain adults, pupae were left in Petri dishes and placed in a 1.125 l Decor Tellfresh plastic container ($12 \times 9.5 \times 10.5$ cm), where adults had access to hydrolysed yeast (MP Biomedicals Cat. no. 02103304), commercial cane sugar (CSR® White Sugar), and water ad libitum upon emergence. 'Percentage of adult emergence' was calculated as the number of adults that emerged divided by the total number of pupae multiplied by 100 (%). To measure adult body mass, recently emerged adults ($n = 9$–10 individuals of each sex per treatment) were allowed to feed for 24 h before being transferred to 15 ml Falcon tubes that were kept at $-20°C$ for 4 h before being weighed using a Sartorius® ME5 balance (0.0001 g precision).

To assess the effects of larval diet treatment on fecundity of resulting adults, groups ($n = 18$) of 10 males and 10 females recently emerged (24 h) per diet treatment per replicate were selected and maintained in a 1.125 l Decor Tellfresh plastic container ($12 \times 9.5 \times 10.5$ cm), with access to hydrolysed yeast, commercial cane sugar and water as described previously. The eggs of all groups were collected for 4 consecutive days when the flies were mature at 10–14 days of age [38] in Petri dishes filled with apple juice (Golden Circle®) and covered with Parafilm that was perforated with a pin (*ca* 15 holes of less than 1 mm) to facilitate oviposition. Apple juice was used to emulate the smell of fruit and attract females to the oviposition device. To assess 'next-generation pupal weight', 50 eggs of each group of adults were sampled when they were 10 and 12 days old (three parental diets, six replicates, two time-points, $n = 36$). Eggs were allowed to develop until pupal stage in the 100% diet. Next-generation pupae were weighed as described above.

Previous studies have shown that diet can modulate the ability of flies to tolerate cold temperatures (e.g. [34,35]). Thus, differences in nutrient concentration in our diet treatments could affect cold tolerance of adult flies. To assess this, eight 1-day-old males and eight 1-day-old females per diet treatment ($n = 48$) were selected and placed individually in 15 ml Falcon tubes. Tubes were then placed in a transparent plastic container of $30 \times 60 \times 30$ cm filled with ice-water maintained at $0°C$ and the latency was measured until flies became immobile at the bottom of the tubes (i.e. 'cold coma', see [39]); the experiment continued until all flies were immobile.

### 3.2.3. Experiment 3: flight ability

Approximately 3500 eggs (250 μl of eggs in suspension) were deposited in 500 ml plastic trays ($17.5 \times 12 \times 4$ cm) containing 150 ml of one of the 100%, 80% or 60% experimental diets with Brewer's yeast and sucrose as described previously (6 replicates/diet, $n = 18$). After 6 days, each tray was placed in a 5 l plastic container with *ca* 150 g of fine vermiculate for pupation. Pupae were sieved from the vermiculate and 100 pupae were sampled from the peak days of collection (days 7 and 8), and weighed as described previously. Pupae were then placed in 90 mm Petri dish lids that were lined with black filter paper. A 100 mm tall acrylic tube (89 mm external diameter $\times$ 84 mm internal diameter) was placed onto the 90 mm Petri dish that contained the pupae. The acrylic tube was painted black on the outside and coated with fine layer of unscented talcum powder on the inside to prevent flies from walking, instead of flying, out of the tube. The tube was placed inside a mesh cage ($32.5 \times 32.5 \times 32.5$ cm, Megaview BugDorm-43030F) under a 20 W fluorescent tube at *ca* 5 cm above the cages, which was on throughout the experiment [37]. To quantify the number of flies that flew out of the tube but later returned and died inside the original tube (flyback), a second empty black tube, also coated with a fine layer of unscented talcum powder, was placed 6 cm away from the tube containing the pupae. By counting the number of flies that flew into the flyback tube, we could estimate the number of flies we would expect to have flown into tube 1. This is important because in tube 1, adult flies could be fliers or non-fliers, and distinguishing them is essential for the correct estimate of flight ability. Finally, flies that flew from tube 1 and remained in the mesh cage were also scored as fliers. We measured pupal weight as described previously. We also measured the sex ratio of emerged adults as the number of fully emerged females divided by the number of fully emerged males. The 'percentage of partially emerged flies' was calculated as the number of adults that attempted emergence but failed to free themselves from the puparium and died divided by the total

number of pupae multiplied by 100 (%). 'Percentage of fliers' was calculated as the number of adult fliers + the number of adults in the flyback tubes + the same number as in the flyback tube in tube 1 (i.e. an estimate of fliers that returned to the original tube), divided by the total number of pupae multiplied by 100 (results in %). 'Rate of fliers' was calculated as the number of adult fliers + the number of adults in the flyback tube + the same number as in the flyback tube in tube 1, divided by the total number of adult emergence (i.e. excluding non-emergence and partial emergence) [40].

## 3.3. Statistical analyses

All statistical analyses and plots were performed in R 3.2.2 [41]. To analyse larval diet choices, we used a multinomial logistic regression (package 'nnet' [42]), which measures the relative odds ratio and the probabilities of choice of a given food patch relative to a reference level (no choice, the agar base between discs of diet was used as the reference level; electronic supplementary material, table S2). We tested the effects of diet treatment on egg hatching success using Kruskal–Wallis test, from which $p$-values were obtained from $\chi^2$-statistics. For pupal recovery, pupal weight, sex ratio, adult body mass and next-generation pupal weight, we assessed statistical significance using two-way ANOVA with time, diet and their interaction as factors. For count data, such as daily pupal recovery and group fecundity, a GLM with Poisson distribution and quasi extension was fitted. For testing group fecundity, female death ($n = 6$ out of 180) was controlled by dividing the total fecundity of the group at a given time by the number of females that contributed to the fecundity of the group at that same time (i.e. 'fecundity per female'). We also controlled for male deaths ($n = 7$ out of 180) in groups by including number of males as a covariate in the models since male harassment can affect female fecundity in other species [18,43]. There were no more than two deaths of either sex in any group. For percentage data, such as percentage of fliers, percentage of adult emergence and percentage of partial emergence, we used a GLM with binomial distribution and quasi extension. For the rate of fliers, we used a GLM with *Gamma* distribution as it provided a better fit of the data. For cold tolerance, we fitted a Cox model (Logrank test = 11.4, d.f. = 5, $p = 0.043$) (package 'survival' [44]) and assessed the significance of sex and diet treatment using $\chi^2$-statistics. GLM $p$-values were obtained through $F$-statistics. We controlled for experimental replicate in all GLM and Cox models (see the electronic supplementary material, tables for complete analyses and model outputs).

# 4. Results

## 4.1. Experiment 1: higher proportions of larvae in patches with higher nutrient concentrations

The largest proportion of larvae was found in patches with 100% nutrient concentration, followed by 80% and 60% nutrient concentration; diets with concentration below 60% were avoided (electronic supplementary material, table S2, and figure S1). There was no evidence of variation in these patterns over time or across foraging densities (i.e. 25, 50 and 100 larvae; figure 1 and electronic supplementary material, figure S1 and table S2), suggesting that the intraspecific competition levels in a foraging location has little effect on larval foraging patterns in nutritionally heterogeneous environments.

## 4.2. Experiment 2: daily pupal recovery and parental pupal weight

We then selected the 100%, 80% and 60% diets from the foraging experiment (i.e. henceforth referred to as 'diet treatment'), which were the range of diets accepted by the foraging larvae across all intraspecific competition levels, and measured egg hatching success as well as pupation traits such as daily pupal recovery and pupal recovery. There was no effect of diet on egg hatching success (Kruskal–Wallis $\chi^2$-value = 4.843, d.f. = 2, $p = 0.088$). Diet treatment did not affect pupal recovery (*Diet treatment*: $F_{2,14} = 1.048$, $p = 0.377$; figure 2a, electronic supplementary material, table S3). However, there were statistically significant effects of diet treatment (*Diet treatment:* $F_{2,50} = 3.818$, $p = 0.030$), as well as the linear and quadratic effects of day after egg collection on daily pupal recovery (*Days after egg collection*: $F_{1,49} = 610.693$  $p < 0.001$, *Days after collection*$^2$: $F_{1,48} = 168.297$, $p < 0.001$; electronic supplementary material, table S3). The significant effect of diet was largely due to a slightly higher rate of pupation on day 7 after egg collection for the diet treatment 60% when compared with other diet treatments (figure 2b). More importantly though was the significant interaction between the linear effect of time and diet treatment on daily pupal recovery (*Days after egg collection*Diet treatment*:

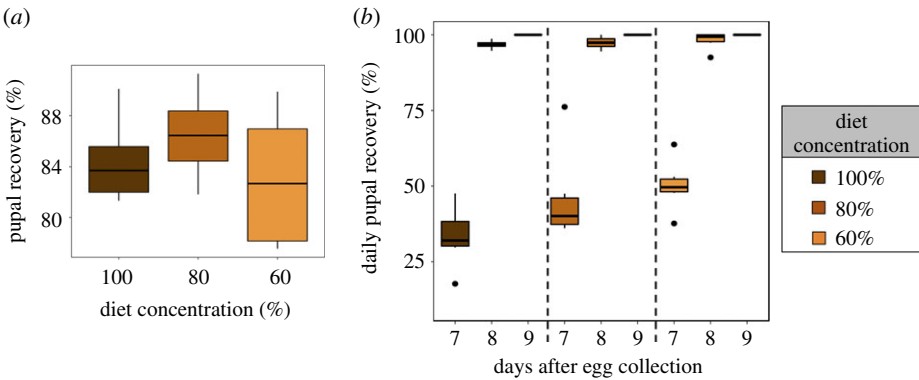

**Figure 2.** The effects of the diet treatment on pupal traits. (*a*) Pupal recovery (%) and (*b*) cumulative daily pupal recovery (%).

$F_{2,44} = 4.994$, $p = 0.001$; electronic supplementary material, table S3), whereby daily pupal recovery increased sharply for diet treatment 100% from day 7 to day 8 after egg collection, but this increase was less evident for diet treatments 80% and 60%, respectively (figure 2*b*, electronic supplementary material, table S3). There was no statistically significant interaction between the quadratic effect of time and diet treatment (electronic supplementary material, table S3). There was also no statistically significant effect of diet treatment on average pupal weight (*Diet treatment*: $F_{2,355} = 0.164$, $p = 0.849$; electronic supplementary material, table S3), and no significant interaction between the day after egg collection and diet treatment on pupal weight (electronic supplementary material, table S3).

## 4.3. Experiment 2: adult emergence and body mass

There was no effect of diet treatment on the percentage of adult emergence (*Diet treatment*: $F_{2,32} = 1.217$, $p = 0.311$; electronic supplementary material, table S4). However, there was a strong effect of time of pupation on the percentage of adult emergence, whereby the percentage of adult emergence was higher for pupae formed on day 8 compared with day 7 (*Day of pupation*: $F_{2,31} = 23.055$, $p < 0.001$, electronic supplementary material, table S4; figure 3*a*). There was no interaction between diet treatment and time of pupation on the percentage of adult emergence (*Diet treatment*Day of pupation*: $F_{2,29} = 0.633$, $p = 0.538$; electronic supplementary material, table S4). We also found no effect of diet treatment on the sex ratio of the emerged adults (electronic supplementary material, table S4). We did find an effect of time of pupation on sex ratio of adults, whereby the sex ratio was female-biased for pupae formed on day 7 but not for pupae formed on day 8 (*Day of pupation*: $F_{2,29} = 14.497$, $p < 0.001$; figure 3*b*, electronic supplementary material, table S4). There was no interaction between diet treatment and time of pupation on sex ratio of emerged adults (*Diet treatment*Day of pupation*: $F_{2,27} = 0.062$, $p = 0.940$; electronic supplementary material, table S4). Together, the results showed that a decrease in macronutrient concentration from 100% of the standard diet to 60% does not influence the viability of the pupae, although adult emergence and the sex ratio of emerged adults can vary with the time of pupation in *B. tryoni*.

We then assessed whether the diet treatments affected adult body mass. Females were heavier than males (mean mass (g) ± s.e.m: *Female*: 15.04 ± 0.155, *Male*: 13.34 ± 0.111, $n = 35$, $F_{1,34} = 78.953$, $p < 0.001$) but there was no effect of diet treatment on either male or female body mass (Male: *Diet treatment*: $F_{2,31} = 0.380$, $p = 0.687$; Female: *Diet treatment*: $F_{2,31} = 0.943$, $p = 0.401$; electronic supplementary material, table S4). There was no effect of time of pupation, or of the interaction between diet treatments and time of pupation on male or female body mass (electronic supplementary material, table S4). These results show decrease in macronutrient concentration from 100% of the standard diet to 60% does not negatively affect adult emergence traits or adult body mass.

## 4.4. Experiment 2: adult cold tolerance

Next, we investigated whether the diet treatments influenced the ability of adult flies to tolerate cold stress. Neither sex nor diet treatment had a significant effect on ability to tolerate low temperature stress (*Sex*: $\chi^2 = 1.755$, d.f. = 1, $p = 0.185$; *Diet treatment*: $\chi^2 = 2.289$, d.f. = 2, $p = 0.242$; electronic supplementary material, figure S2 and table S5).

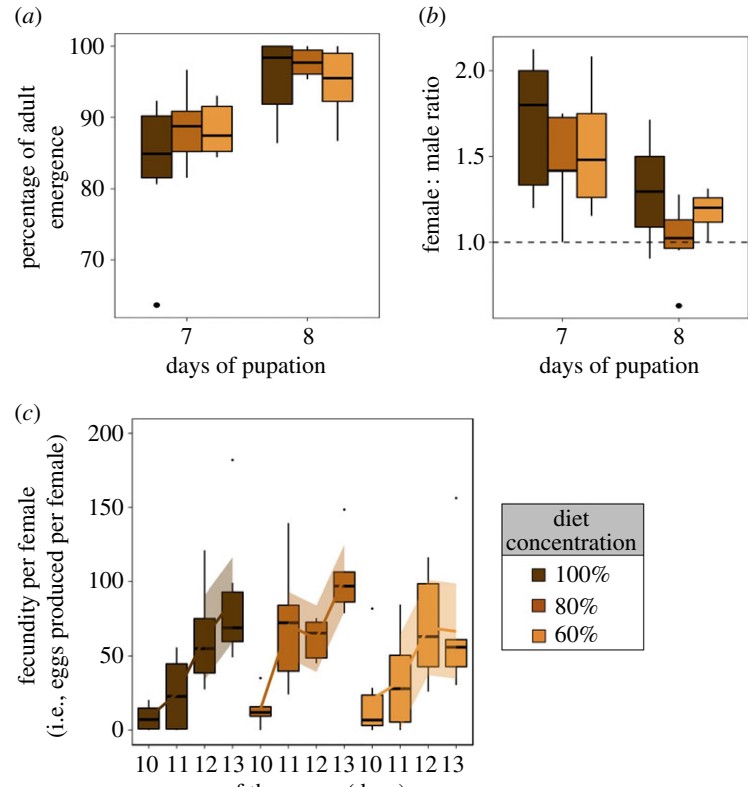

**Figure 3.** The effects of the diet treatment on adult traits. (*a*) Percentage of adult emergence (%). (*b*) Sex ratio (female : male) of emerged adults. Note that day of pupation corresponds to the day in which larvae pupated after egg collection. (*c*) Fecundity per female (i.e. number of eggs per females in the group) over 4 consecutive days, from the first day of sexual maturity of the group (day 10). Lines were made in ggplot2 with the 'loess' method and were drawn to show the trend in our data [45].

## 4.5. Experiment 2: reproduction and next-generation pupal weight

We also investigated whether larval diet dilution affected adult reproduction or weight of pupae in the next generation. Fecundity per female increased over time (*Age of the group*: $F_{2,66} = 40.625$, $p < 0.001$; figure 3*c*, electronic supplementary material, table S6). However, no effect of diet treatment on per female fecundity was detected (*Diet treatment*: $F_{2,67} = 2.210$, $p = 0.118$; electronic supplementary material, table S6). The nonlinear effects of the age of the group, as well as the interactions between the linear and nonlinear effects of the age of the adult group and diet treatment were not significant (electronic supplementary material, table S6). Moreover, there was no significant effect of diet treatment on the total fecundity of the adult groups (i.e. total number of eggs produced over the 4-day period) (*Diet treatment*: $F_{2,14} = 1.430$, $p = 0.272$; electronic supplementary material, table S6). We then measured next-generation pupal weight, but found no effect of diet treatment or age of the adult group on next-generation pupal weight (electronic supplementary material, table S6). There was also no significant interaction between the age and the diet treatment of the adult group as predictors of next-generation pupal weight (electronic supplementary material, table S6).

## 4.6. Experiment 3: adult flight ability

We then investigated whether the diet treatments influenced the general performance of adult in their ability to fly. We confirmed our previous results that macronutrient dilution of the larval diet had no effect on the average weight of the pupae, the percentage of partially emerged flies, percentage of adult emergence and the sex ratio of emerged adults (see electronic supplementary material, table S7). There was also no significant effect of diet treatment on the percentage of fliers (*Diet treatment*: $F_{2,14} = 1.446$, $p = 0.264$; electronic supplementary material, table S7) or on the rate of fliers (*Diet treatment*: $F_{2,14} = 1.197$, $p = 0.331$; electronic supplementary material, table S7), suggesting that the diet treatments did not affect the ability of adults to fly.

# 5. Discussion

The foraging behaviour of insect larvae is an important factor underlying individual development, growth and fitness. In species that encounter complex or diverse nutritional environments, such as polyphagous fruit flies, heterogeneity in the diet substrate may induce variation in larval nutrition that can canalize larval development and lead to downstream fitness effects in adulthood. Here we used larval foraging choices in a competitive context as a guide to test whether larvae make diet choices that could sustain growth and fitness, independent of intraspecific competition levels. We predicted that larvae would prefer diets with the highest nutrients when given a choice (prediction 1), irrespective of the density of conspecifics in the foraging arena (prediction 2). When given diets chosen in the foraging experiment throughout egg-to-adult development, we predicted that diets with lower nutrient dilution would result in adults with lower body mass and fecundity (prediction 3), although we did not have any *a priori* knowledge from previous studies to expect an effect of diet on flight ability and stress tolerance (prediction 4). Our data indicate that the range of dilute diet patches accepted by *B. tryoni* larvae when foraging in a competitive nutritionally heterogeneous environment can support larval development, as well as adult and offspring traits, including adult body mass and fecundity. For instance, *B. tryoni* larvae consistently accepted diets ranging from 60% to 100% concentration, with more larvae feeding on more concentrated diets—a pattern consistent across all larval densities. When allowed to develop from egg to adult in these diets, those on diets of 100 and 80% concentrations displayed an increase in daily pupal recovery compared with diet with 60% concentration, but there was no evidence of differences among diets of 100%, 80% or 60% in pupal recovery, pupal weight or fitness-related traits such as adult body mass, flight ability, fecundity and tolerance of cold temperatures. Furthermore, no evidence of trans-generational effect on offspring pupal weight was found for any of the diet treatments. Taken together, these results suggest that *B. tryoni* larvae are able to accommodate diets that support development and adult fitness.

Previous studies have demonstrated that larval density can modulate larval foraging behaviour [46–48], although we found no influence of larval competition levels on larval foraging choices in heterogeneous nutritional environments. More importantly though, the degree of environmental heterogeneity is an important factor modulating foraging behaviour and the expression of life-history traits [49,50]. Here, when we investigated how larval choices influenced life-history traits, we found no effects of diet dilution (which were based on the larval feeding choices) on any fitness-related traits of current and next generations assessed in our study apart from a period after pupation. These findings suggest that the polyphagous fruit fly *B. tryoni* is resilient to mild diet quality fluctuations, which may be an important factor contributing to their success in exploiting a variety of fruit resources (*B. tryoni* is a major horticulture pest in Australia [51,52]). Thus, the findings of our study can potentially shed light into how polyphagous species are able to colonize and thrive in changing and spatially variable environments. More studies are needed on how nutritional heterogeneity affects fitness of monophagous and polyphagous species in order to draw general principles of how some species can expand their geographical distribution while others do not.

While no effect was found on most fitness traits, we did find that nutrient concentration significantly affected daily pupal recovery. This result corroborates previous findings in other tephritid species as well as other flies showing that the composition of the larval diet can delay larval development. For instance, in medflies (*Ceratitis capitata*), low-protein diets delay larval development and adult maturation (e.g. [11]). In *Bactrocera dorsalis* and in *B. tryoni*, larval development is delayed on larval diets with certain combinations or amounts of both macro- and micronutrient, as well as with some physical formulations of the diet (e.g. gel, liquid; e.g. [37,53,54]). In Drosophilids, diets with certain nutrient compositions have also been reported to delay larval development [23,55]. While the delays in development caused by diet quality have been largely interpreted as effects on larval growth rate *per se*, the results of these studies might reflect at least in part the diet-dependent differences in the timing of egg hatching, which are carried over through development. For instance, in other insects, the nutritional properties of the oviposition site can impact the timing of egg hatching [20,21,25,56], but the mechanisms are still unknown [57]. We did not find an effect of nutrient concentration on egg hatching success, but we cannot rule out the possibility that diet affected egg hatching timing. It is also possible that feeding compensatory mechanisms played a role in generating the patterns observed in our study, but this remains subject of future investigation. It is important to mention that external conditions (e.g. temperature, humidity) in our experiments were controlled. Had external factors fluctuated and/or were suboptimal, the delays in daily pupal recovery could have been more accentuated (e.g. [58]). Future studies addressing environment-dependent expression of life-history

traits in *B. tryoni* are needed to test whether external factors can promote costs of foraging behaviour in competitive heterogeneous nutritional environments.

Numerous previous studies of other insects have reported that the developmental diet is a potent modulator of adult traits (e.g. [5,7–12]). In this study, however, no significant effect of diet was detected on larval or adult fitness-related traits. The discrepancy in results is probably due to differences in the experimental approach. Whereas in previous studies larvae were provided diets with often extreme nutrient dilutions or composition, beyond the range that would be accepted in a choice setting, in the present study larvae were provided dilutions that were based on the range they accepted in a multi-choice foraging assay. As a result, our diet dilutions would probably have caused much less severe dietary restriction than those imposed by other studies. Note that in our study the larval foraging patterns observed could have emerged due to stable individual variation among larvae in preferred nutrient concentrations such that the largest proportion prefers the highest concentration (100%), and smaller proportions prefer lower concentrations (80%, and then 60%). Also, there could be within-individual variation such as, for example, if some larvae foraged only in 100% and 80% diets, whereas other forage in 100%, 80% and 60%. It is also possible that larvae are mobile and spend more time handling high concentration food, which would in turn induce the formation of larger larval aggregations on patches of higher nutrient concentration. The mechanisms underlying individual larval foraging behaviour and dietary preferences are unknown in *B. tryoni* at this time, and this remains an important topic for future investigations.

## 6. Conclusion

The present study reveals that within the range of diet dilutions accepted by polyphagous larvae under choice conditions, there was no evidence of variation among diets in developmental, adult or next-generation fitness-related traits following development from eggs to adulthood on these diets. An ability to accommodate variation in competition levels and nutrient concentration may be important for development and fitness, especially for larvae from polyphagous species such as *B. tryoni*, which often encounter unpredictably diverse nutritional and social environments when infesting a wide variety of host fruits across its geographical distribution.

Data accessibility. Data available from the Dryad Digital Repository at: https://doi.org/10.5061/dryad.35h2v0b [59].
Authors' contributions. J.M. and F.P. designed the experiment. J.M., S.T.T., I.L. and B.M. collected the data. J.M., P.W.T. and F.P. analysed the data. All authors contributed to the writing of the manuscript.
Competing interests. The authors have no competing interests to declare.
Funding. This research was conducted as part of the SITplus collaborative fruit fly programme. Project Raising Q-fly Sterile Insect Technique to World Standard (HG14033) is funded by the Hort Frontiers Fruit Fly Fund, part of the Hort Frontiers strategic partnership initiative developed by Hort Innovation, with co-investment from Macquarie University and contributions from the Australian Government.

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
