## [Reviewer comments · Royal Society Open Science]

Review History

RSOS-190090.R0 (Original submission)

Review form: Reviewer 1

Is the manuscript scientifically sound in its present form?

Yes

Are the interpretations and conclusions justified by the results?

Yes

Is the language acceptable?

Yes

Is it clear how to access all supporting data?

Yes

Do you have any ethical concerns with this paper?

No

Have you any concerns about statistical analyses in this paper?

No

Recommendation?

Accept with minor revision (please list in comments)

Comments to the Author(s)

The revised version is now somewhat clearer. A few points remain to be dealt with:

1. As mentioned in my first review, I would prefer to see the hypotheses the authors had about the outcome of the study at the end of the introduction! In the discussion the authors should come back to these expectations!
2. As also mentioned last time, the authors highlight the importance of studying fly responses under different larval competition levels. However, larval competition was only manipulated in experiment 1, which lasted only for 4 h, but density was kept constant in the other experiments, which investigated performance over longer time, where differences due to larval density may have been more likely. Thus, the text should be rephrased accordingly (e.g. abstract, line 35ff and lines 93ff), making clear that only very short-time competition was in fact investigated!
3. "Nutrient concentration" and "diet concentration" are now both used. Throughout the manuscript, only one of these terms should be used.
4. Sample size should be also given in the Tables of the Supplement, not only in the text.

Decision letter (RSOS-190090.R0)

07-Mar-2019

Dear Dr Morimoto

On behalf of the Editors, I am pleased to inform you that your Manuscript RSOS-190090 entitled "Larval foraging decisions in competitive heterogeneous environments accommodate diets that support egg-to-adult development in a polyphagous fly" has been accepted for publication in Royal Society Open Science subject to minor revision in accordance with the referee suggestions. Please find the referees' comments at the end of this email.

The reviewers and handling editors have recommended publication, but also suggest some minor revisions to your manuscript. Therefore, I invite you to respond to the comments and revise your manuscript.

- Ethics statement

- Data accessibility

It is a condition of publication that all supporting data are made available either as supplementary information or preferably in a suitable permanent repository. The data

accessibility section should state where the article's supporting data can be accessed. This section should also include details, where possible of where to access other relevant research materials such as statistical tools, protocols, software etc can be accessed. If the data has been deposited in an external repository this section should list the database, accession number and link to the DOI for all data from the article that has been made publicly available. Data sets that have been deposited in an external repository and have a DOI should also be appropriately cited in the manuscript and included in the reference list.

If you wish to submit your supporting data or code to Dryad (<http://datadryad.org/>), or modify your current submission to dryad, please use the following link:
<http://datadryad.org/submit?journalID=RSOS&manu=RSOS-190090>

- **Competing interests**

- **Authors' contributions**

- **Acknowledgements**

- **Funding statement**

Because the schedule for publication is very tight, it is a condition of publication that you submit the revised version of your manuscript before 16-Mar-2019. Please note that the revision deadline will expire at 00.00am on this date. If you do not think you will be able to meet this date please let me know immediately.

To revise your manuscript, log into <https://mc.manuscriptcentral.com/rsos> and enter your Author Centre, where you will find your manuscript title listed under "Manuscripts with

Decisions". Under "Actions," click on "Create a Revision." You will be unable to make your revisions on the originally submitted version of the manuscript. Instead, revise your manuscript and upload a new version through your Author Centre.

on behalf of Prof Kevin Padian (Subject Editor)
openscience@royalsociety.org

Associate Editor Comments to Author:

The reviewer considers that their requests have been largely tackled successfully, though a few items remain before the paper may be accepted for publication. Please ensure you fully address these in your revision.

Reviewer comments to Author:

Reviewer: 1

Comments to the Author(s)

The revised version is now somewhat clearer. A few points remain to be dealt with:

1. As mentioned in my first review, I would prefer to see the hypotheses the authors had about the outcome of the study at the end of the introduction! In the discussion the authors should come back to these expectations!
2. As also mentioned last time, the authors highlight the importance of studying fly responses under different larval competition levels. However, larval competition was only manipulated in experiment 1, which lasted only for 4 h, but density was kept constant in the other experiments, which investigated performance over longer time, where differences due to larval density may have been more likely. Thus, the text should be rephrased accordingly (e.g. abstract, line 35ff and lines 93ff), making clear that only very short-time competition was in fact investigated!
3. "Nutrient concentration" and "diet concentration" are now both used. Throughout the manuscript, only one of these terms should be used.
4. Sample size should be also given in the Tables of the Supplement, not only in the text.

Author's Response to Decision Letter for (RSOS-190090.R0)

See Appendix A.

Decision letter (RSOS-190090.R1)

20-Mar-2019

Dear Dr Morimoto,

I am pleased to inform you that your manuscript entitled "Larval foraging decisions in competitive heterogeneous environments accommodate diets that support egg-to-adult development in a polyphagous fly" is now accepted for publication in Royal Society Open Science.

on behalf of Mr Andrew Dunn (Associate Editor) and Kevin Padian (Subject Editor)
openscience@royalsociety.org

Associate Editor Comments to Author (Mr Andrew Dunn):

Associate Editor: 1

Comments to the Author:

(There are no comments.)

Reviewer comments to Author:

Follow Royal Society Publishing on Twitter: [@RSocPublishing](https://twitter.com/RSocPublishing)

Appendix A

Morimoto *et al.* "Larval foraging decisions in competitive heterogeneous environments accommodate diets that support egg-to-adult development in a polyphagous fly"

We sincerely thank the editor the reviewers for their positive assessment of our manuscript. To address the reviewers' comments, we:

- Added predictions at the end of the Introduction, and reiterate our predictions in the first paragraph of the Discussion;
- Clarified the experimental design and the scope of our inferences, as well as the use of terms throughout the text;
- Included sample sizes in tables in which degrees of freedom were not previously provided.
- We also corrected the acknowledgment statement as requested by our funding agency.

Please find a complete point-by-point response to the reviewer's comments below (lines correspond to the file with track changes).

Royal Society Open Science - Decision on Manuscript ID RSOS-190090 Associate Editor

The reviewer considers that their requests have been largely tackled successfully, though a few items remain before the paper may be accepted for publication. Please ensure you fully address these in your revision.

Reviewer(s)' Comments to Author:

Referee: 1

Comments to the Author(s)

The revised version is now somewhat clearer. A few points remain to be dealt with:

1. *As mentioned in my first review, I would prefer to see the hypotheses the authors had about the outcome of the study at the end of the introduction! In the discussion the authors should come back to these expectations!*

Response: We have now included predictions at the end of the introduction (Ln 117-130) and adjusted the discussion to reiterate the predictions before discussing the broad relevance of our findings (Ln 401-408).

2. *As also mentioned last time, the authors highlight the importance of studying fly responses under different larval competition levels. However, larval competition was only manipulated in experiment 1, which lasted only for 4 h, but density was kept constant in the other*

experiments, which investigated performance over longer time, where differences due to larval density may have been more likely. Thus, the text should be rephrased accordingly (e.g. abstract, line 35ff and lines 93ff), making clear that only very short-time competition was in fact investigated!

Response: We apologise for this. We have now clarified the text to mention that larval competition was only manipulated in a short-time larval choice experiment (see Ln 38-39, 95, 109).

- 3. “Nutrient concentration” and “diet concentration” are now both used. Throughout the manuscript, only one of these terms should be used.*

Response: We have now corrected this (e.g., Ln 300, 302-303).

- 4. Sample size should be also given in the Tables of the Supplement, not only in the text.*

Response: We have now included the sample size information for Tables S2 and S5 in the table’s caption (see Revised Supplementary Material). For all other supplementary tables, the sample size information is in the degrees of freedom within each table.